

**Dust Opacities inside Dust Devil Column in the Taklimakan Desert**
Zhaopeng Luan[1,2,3], Yongxiang Han[1,2], Feng Liu[1,2], Tianliang Zhao[1,2] , Chong Liu[1,2],
Mark J. Rood[4], Xinghua Yang[5], Qing He[5], Huichao Lu[3]
[1] Collaborative Innovation Center on Forecast and Evaluation of Meteorological Disasters,
Nanjing University of Information Science and Technology, Nanjing, 210044, China
[2] Key Laboratory for Aerosol-Cloud-Precipitation of China Meteorological Administration,
Nanjing University of Information Science and Technology, Nanjing, 210044, China
[3] Tai'an Meteorological Brueau, Tai'an, 271000, China
[4] Department of Civil & Environmental Engineering, University of Illinois at Urbana-Champaign,
Illinois ,61820, USA
[5] Institute of Desert Meteorology, China Meteorological Administration, Urumqi ,830002, China
Correspondence to: Tianliang Zhao (tlzhao@nuist.edu.cn)
**Abstract**. The distribution of particulate matter (dust) in dust devils (DDs), which are
well-defined vortexes of wind that range from 1 m to 1,000 m tall, is quantitatively
quantified here based on light transmission. We applied the Digital Optical Method
(DOM) with digital still cameras to quantify the opacity of the DDs in the Taklimakan
Desert, China. This study presents the following unique and important results and
interpretations: 1) the distinct horizontal distributions of opacity indirectly proved the
existence of DDs' zone of weak winds in the center of the swirling vortex, similar like
the eye of tropical cyclone, which is difficult to be observed directly; 2) The opacity
of the DDs decreases with increasing height, however, the dust does appears to settle
out, and the relatively calm eye leads to a minimum in dust opacity at the eye; 3) The





horizontal distribution of opacity is quasi symmetric with a bimodal across the eye of
the DDs; and 4) A new robust method is developed for the representation of
three-dimensional structure of opacity based on the observed two-dimensional
structures provided by DOM. This research not only proposes a highly reliable, low
cost and efficient methodology to characterize the inside structure of DDs, but also
provides the information on estimation of dust emissions caused by DDs.
**Keywords**: dust devil; digital optical; opacity; dust devil eye; 3-D structure

## 1 Introduction

Well-defined vortexes of wind that range from 1 m to 1,000 m tall, also known as
dust devils (DDs) are the most common small scale (< 50 m diameter at ground level)
dust transmitting system in the atmosphere(Kanak, 2005;Kanak et al., 2000;Leovy,
2003). It is a special case of columnar, ground-based convective vortex occurring in
the lower atmospheric boundary layer (Gu et al., 2008;Koch and Renno, 2005).
Occurrences of wind devils are associated with weak wind, sunny weather, and local
surface pressure fluctuations which are dominated by heterogeneous solar radiation
due to uneven heating of the ground, that leads to a rising convection vortex rotation
containing particulate matter (PM, dust) under certain conditions of angular
momentum(Duan et al., 2013).The theory and numerical model results suggest that
DDs have significant potential for high dust loadings. DDs contribute to one third of
the total natural particulate mass emitted to the atmosphere annually (Koch and Renno,
2005). Emissions of total primary particulate mass by DDs to the atmosphere is as




high as 65% in the United States (Gillette and Sinclair, 1990). About 77%-87% of the
primary particulate mass emitted to the atmosphere in Chinese deserts is caused by
DDs (Han et al., 2008;Deng et al., 2011;Zhang et al., 1994), while DDs generate 30%
of the primary particulate mass emitted to the atmosphere in the Sahara Desert
(Marsham et al., 2008). These results imply that DDs may play an equal or more
important role than sandstorms when considering the total emissions of PM into the
atmosphere. DDs affect not only air quality of arid areas but also impact global or
regional climate through the "umbrella" (Wei et al., 1998) and "condensation nucleus"
effects (Wang and Zhang, 2001) in the atmosphere as well as the "iron fertilizer effect"
in open oceans (Han et al., 2006) that occur because of the remote entrainment,
transmission and then deposition of dust from the atmosphere.
However, physical processes and mechanisms of dust entrainment by DDs are
poorly understood. It is extremely difficult to observe DDs because of their sporadic
and unpredictable mobile paths. So the basic structure and characteristic parameters
have long been based on human observations by Ives (1947) and Sinclair (1964). DDs'
central core structure has been remarkably observed with mobile Doppler radar
(Bluestein et al., 2004) and their center pressure has been characterized by pressure
logger (Lorenz, 2012;Lorenz, 2013). However, data obtained from the mobile
Doppler radar and pressure logger depend on locating the instruments along the
estimated path of the DDs where they may appear, and the measurements are mainly
focused on meteorological parameters. The high-cost measurement implementation
requires extensive and expensive resources. In addition, LES(Large Eddy Simulation)





have also been applied to DD studies by Toigo (2003), Ito et al.(2010) and Kanak et al.
(2000; 2005). The condition of weaker wind and stronger surface heat flux favorable
for the formation of dust devil is confirmed according to theirs studies. Gu et al. (2003;
2010) even provided the meteorological characteristics inside of DDs by numerical
simulation. However, field observations of DDs to compare to and validate the
modeled results including vortex dynamics are critical and necessary (Leovy, 2003).
The Digital Optical Method (DOM) was developed to measure opacities of plumes
in the atmosphere emitted from stationary point sources (Du et al., 2007; Du et al.,
2009). DOM uses a digital still camera and software for processing the digital pictures
to determine plume opacity. This method was also used to quantify plume opacities
for fugitive dust plumes in the atmosphere (Du et al., 2013). DOM was developed to
quantify plume opacity at low cost, easy implementation, with improved accuracy
compared to human observations of plume opacity, and to provide digital record.
We conducted a DD field campaign in the Taklimakan Desert, the largest desert in
China and the world's second largest liquidity desert during July 2014. This is the first
time to use DOM to observe DDs. Two dimensional (2D) structures of dust devils are
archived with the digital images of the DD and processing with DOM software. This
effort provides horizontal and vertical distributions of opacity values in DDs, and
presents the first three-dimensional (3D) opacity structure of DDs. It is important to
quantify the structure which is helpful to estimate dust emissions of DDs. The study
thus services to provide insightful information for DD numerical simulation and
validation.



## 2 Methods

*2.1 Image acquisition and processing*

The observational investigation site is located in the Xiaotang region (40°50′N,84°10′E,altitude 943.9m), in the hinterland location of the Taklimakan Desert (see Fig.1). Xiaotang is a typical desert-Gobi transitional zone where DDs occur frequently according to local meteorological observations. During the observations from 2 to 14 July, 2014 digital images were obtained with two digital still cameras (Sony Cyber-shot Model DSC-P100). In order to obtain the appropriate background for taking the pictures, the camera is back to the sun within a 140° sector (ASTM D-7520). All pictures archived as JPEG files. Most observed DDs show inverted circular cone and are quasi-symmetric shape across the DD's eye with a height of several tens to hundreds meters. None of the DDs have absolute symmetry due to the heterogeneous distribution of dust particles in the DD caused by the time-dependent dynamics.

A digital image of a typical DD is illustrated in Fig.1 to describe how to determine the DD opacity values. The vertical curved lines were added to the image to represent auxiliary lines describing vertical grids for estimation of opacity values. The sky background of the DD is relatively uniform as shown in Fig.1. The upright electric pole right next to the DD with a height of 4.5 m was a reference to measure the height and diameter of the DDs. The DD with a base radius of 6.4 m and a height of 23.0 m has a maximum diameter of 15.8 m at the height of 16.0 m. In order to examine the




spatial distribution of dust concentration within the DD's dust cone, we horizontally
set the cone into 0.4 m grids with 15 auxiliary upward lines parallel to the DD's
conical surface. The centerline (line 0) is a vertical line from the bottom of the DD's
eye. For the sake of clarity, we count lines on the left of the centerline from 1 to 7,
while denoting lines to the right of the center as A to H (Fig.1b).
*2.2 Calculation of opacity*
DOM's transmission model was used in this study due to the uniform background
sky conditions (e.g., uniform clear or overcast sky) for the DDs. And the challenge to
install contrasting backgrounds behind and next to the DDs as needed with DOM's
contrast model (Du et al., 2007). The transmission model determines the plume's
opacity based on the radiance from the plume and the radiance from the plume's
background. Part of the radiance ($N_0$) from the sky is lost as it passes through the
plume due to light scattering and/or absorption. N denotes the radiances received by
the charge-coupled device (CCD) of the digital camera that correspond to the sky
background, in terms of pixel values:

$$N = N_0 T_0 + N^*$$

$T_0$ denotes the transmittance of the plume-free atmosphere along the path between the
camera and the sky background. $N^*$ is the path radiance of the atmosphere along the
same path , which is from direct ,diffuse, and reflected radiances scattered into the
sight path by ambient air and aerosols (Du et al., 2013), and can be estimated with an
equilibrium radiance model for uniform illumination and negligible absorption:
$$N^* = N_0(1 - T_0)$$



From the two equation, getting the results of $N = N_0$.

2        Therefore, when the radiance reaches the camera ($N_p$), it is caused by the attenuated

radiance value from the plume ($N_{T1}$), diffusive radiance value ($N_{T2}$), and attenuation
of the radiance caused by the surrounding atmosphere (which is negligible compared
to the DD). The attenuated radiance value ($N_{T1}$) results from $N_0$ after the light is
scattered and/or absorbed by the plume and the diffusive radiance value ($N_{T2}$) is
caused by other sources of light than the uniform sky background (Fig.2). According
to the definition of opacity:

$$Opacity = 1 - \frac{N_{t1}}{N_0} = 1 - \frac{N_p - N_{t2}}{N}$$

Because the camera did not directly measure $N_{t2}$, the proportionality coefficient, K, is
defined by $N_{t2} = K*N*Opacity$ (Du et al., 2007), and then the plume opacity could be
determined by the transmission model as described by:

$$Opacity = \frac{1 - \frac{N_p}{N}}{1 - K}$$

$N_p$ is the equivalent radiance value recorded by the camera, in terms of pixel values,
caused by radiance from the plume ($N_T$) and path radiance of the atmosphere. N is the
equivalent radiance value recorded by the camera, in terms of pixel values, after $N_0$
passes through the DD-free atmosphere. K value of 1.4 in the transmission model is
used.
**3 Results and discussion**
*3.1 The vertical profile of opacity*

21        Fig. 3a, b, and c demonstrate the opacity profiles of the vertical grid lines, as shown



in Fig.1. It can be seen that: 1) The opacity profiles are non-uniform distributions in
the vertical direction, and the vertical variation of opacity becomes more significant
from the inner to the outer portions of the conical dust plume (See from the both
Fig.3a and b), 2) The opacity profiles from center to both sides show quasi-symmetry
but none of them are absolutely symmetrical; and 3) The opacity values generally
decrease with height. However, the opacity decreases more rapidly at lower levels
than upper ones. The averaged lapse rate (percent change in opacity values with
increasing height) for the profile is 4.1% per meter below 10 m height over which the
lapse rate drops to 0.6% per meter (Fig.3c).
*3.2 The horizontal variation of opacity*
From the horizontal variation of the opacity at different levels, it is observed that: 1)
the horizontal opacity values increase first and then decrease from the center part to
both sides at the base of the DD, implying that the DD is quasi-symmetric; 2) The
opacity caused by dust aerosols and the corresponding dust concentrations are
decreasing with height within the DD, and the opacity is a monotonically decreasing
function as height increase. The maximum opacity is observed at the bottom of the
DD (Fig. 4a); and 3) on the right side of the DD, the magnitude of opacity is greater
than the left side (Fig.4). Therefore DOM method is able to capture all important
well-known natures of DD including non-absolute symmetry and internal
inhomogeneity.
*3.3 Two-dimensional distribution of the quantized DD's opacity*
2D grid boxes shown in Fig. 5 describe the more detailed 2D spatial distributions of





the opacity values. Each grid box with a unique averaged opacity value corresponds to
a specific dust concentration. The 2D distribution image reflects the basic features of
inhomogeneity and quasi-symmetry inside the DD. The grids used to quantify spatial
distribution of opacity values are able to present the similar spatial variability like grid
boxes defined in DD numerical models to simulate spatial distribution of physical
properties(Mason et al., 2013;Gu et al., 2006).

7       The opacity is related to the PM's concentration, composition, and size distribution

in DDs (Metzger et al., 1999). The value of the opacity is assumed to decrease with
increasing height due to gravity that prevents larger particles from traveling to the
upper parts of the DD. Therefore, it is expected that smaller particles could be
transmitted to a higher altitud(Gu et al., 2003;Gu et al., 2007). A distinct vertical
opacity gradient develops with small values at the center portion of the DD. It
suggests that airflow inside DD is relatively stable and vertical mixing is weak. At the
same time larger particles are transported into the outer spiral dust bands where the
mixing is much stronger because of entrainment. In addition, the large number of sand
vortexes at lower portions of spiral bands explains the coexistence of descending
larger particles and ascending fine particles. And these results are consistent with
those from numerical simulations by Gu (2007) and Gierasch (1973; 1974).
*3.4 The DD eye and the three dimensional distribution of opacity*
Previous research have clarified the principal characteristics of DDs that the centers
of DDs have low pressure, weak airflow, and almost zero tangential velocity (Fiedler
and Kanak, 2001;Kanak, 2006). Dust concentration in this inner core region is much



lower compared to other portion of DD (Balme and Greeley, 2006;Gu et al., 2003). It
is similar to the eye of hurricanes or typhoons characterized by light wind, clear skies.
Though the DD's eye is difficult to observe directly, the variation of opacity with a
bimodal distribution at the same height(Fig.4)indicates that the DD's eye does exist.
Fig. 6 illustrates a conceptual mode of the horizontal cross section of a DD. The DD
DD's eye diameters are measured by 2R and 2r, respectively, The line segment FG is
the distance between two peak opacity points (F and G) away from the eye, H with
minimum opacity is the center of the eye. The opacity value at point A in Fig. 6, for
instance, is the accumulated opacity through distance from B to C. The changes in
opacity from B to C, however, is unknown. So, we assume the horizontal opacity
values are constant between B and C.
A more detailed example illustrated by Fig. 4 describes the DD having a diameter
of 12 m and a height of 4 m. The horizontal variation of opacity values shows an
obvious bimodal distribution. The peak opacity value is located at 3 m away from the
DD's centerline. Therefore, it can be determined that the outer radius of DD is 6 m
and the inner radius of DD's eye is 3 m at the same height. Every segment (e.g., B to
C) is divided into four equal sub-segments, and the horizontal distribution can be
derived accordingly. With this method, we can obtain a 3D structure of the DD by
calculating the horizontal distribution at different heights as shown in Fig. 7 (i.e., at
3m, 6 m, 9 m, and 12 m). The opacity values increase first and then decreases from
the center to both sides of the DD that make up the bimodal distribution. The 3D
structure of DD as described in Fig. 7 represents the fundamental features of





inhomogeneity and quasi-symmetric opacity distribution.

2        The results demonstrate that the methodology to derive the 3D opacity structure in

a DD is feasible and practicable. The future effort to improve this methodology is to
identify and quantify discrepancies between the assumed opacity values and observed
ones from B to C. It can be realized using more than one camera to provide digital
images of the same DD from different angles. The more precise and accurate 3D
structure of DD will be obtained with the improved measurements.
**4 Summary and Conclusions**

10       This study is the first to apply Digital Optical Method (DOM) to quantify the

opacity of dust devils (DDs) in the Taklimakan Desert. The two dimensional (2D)
distribution of opacity values inside the DD was obtained by DOM and correspondent
observational data analysis. Analysis results show that the opacity values of DDs
decrease monotonically with height; the opacity decreases more significantly at lower
levels than upper ones. Also, the averaged lapse rate for the profile is 4.1% below 10
m height over which the lapse rate drops to 0.6%. The horizontal opacity values from
the DD's eye to both edges of the DD increases, reaches a peak value, and then
diminishes rapidly. The quasi-asymmetry distribution of dust particles inside DDs can
be characterized by the bimodal distribution from observed opacity values from the
DD's eye to the outward horizontal directions. The distinct horizontal distribution of
opacity values proves the existence of the DD's eye which is difficult to be observed
directly. A typical 3D structure of opacities is developed based on the assumption of





constant opacity through horizontal cross-sections of a DD. Such results are
encouraging and support the use of DOM as alternative to other instrumentally
intensive, generally expensive methodologies for DD observations.
**5 Prospect**
In order to improve the quality of DD observations using DOM, the following key
issues will be addressed: 1) the proportionality coefficient (K) is not available so far
for desert dust devils. The K value we used in our opacity calculations is for white
plume (light scattering plume) whose optical properties is definitely different form
DD (desert dust particles), the estimate of K for desert dust is identified as a urgent
need for our future study; 2) a scaling model of DD size from DD images is required
if there is no reference to determine DD scale; 3) the results documented in this study
are far from generalized characteristics of DD opacity. In order to realize statistical
analysis of large sized samples, more observations including carefully-designed
specific field campaigns are needed for characterizing DD opacities; 4) the validation
of calculated opacities inside DD column is also a challenge for us. It is essential to
collect other independent observation data of DD for this purpose.
A new method to convert 2D structure of opacity using multiple-camera digital
optical method (MDOM) to 3D will be developed. The results from further improved
MDOM will be related to the estimate of vertical gradient of dust concentration. In
addition, the wind shear and Reynolds stress will be considered in the improved
system in order to realize the parameterization of vertical dust flux caused by DDs.
The opacity of dust devils may also be affected by meteorology like ambient wind



and temperature difference between surface and air. Which may influence the strength
and size of dust devils. High-strength dust devils can roll up more dust particles,
affecting the opacity of dust devils. So in the following study, more observation sites
will be set to measure meteorology such as ambient wind, temperature difference
between surface and air, wind of dust devils, temperature and pressure difference
between core of dust devils and surrounding environment, etc.

## Acknowledgments

This research was supported the National Natural Science Foundation of China
(41375158; 41175093; 41405013) and the Research Starting Project of NUIST
(20110304) of China. Any opinions, findings, and conclusions or recommendations
expressed in this paper are those of the authors and do not necessarily reflect the view
of the Nanjing University of Information Science and Technology, University of
Illinois at Urban and Champaign, or any organizations and government agencies.

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

3    English abstract)

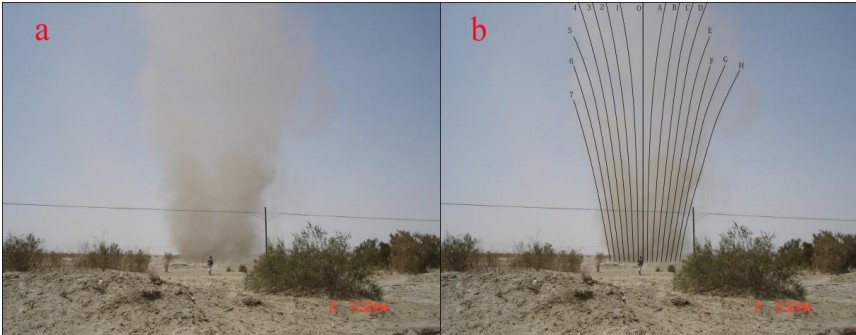

11    **Fig.1.** A typical DD observed on July 2, 2014 (a is the original figure, b is the figure with

12    centerline (line 0) and guides. An upright pole near the DD as a reference for measuring the DD)

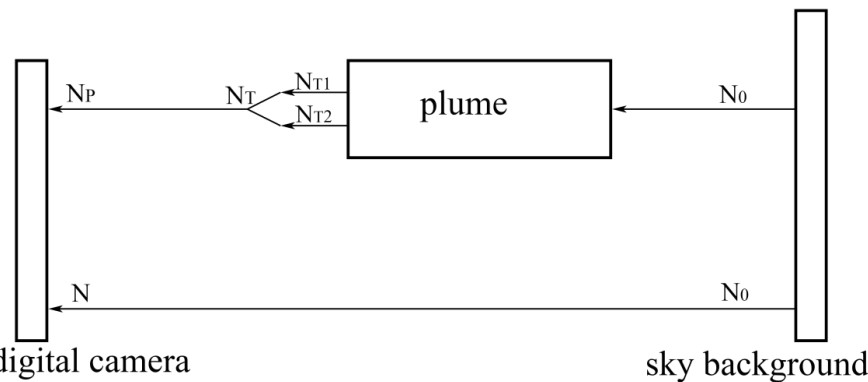

15    **Fig.2.** Schematic describing the transmission model to determine plume opacity





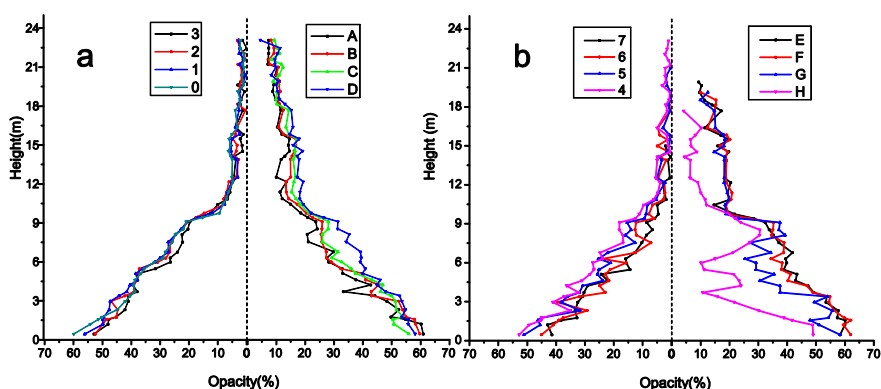

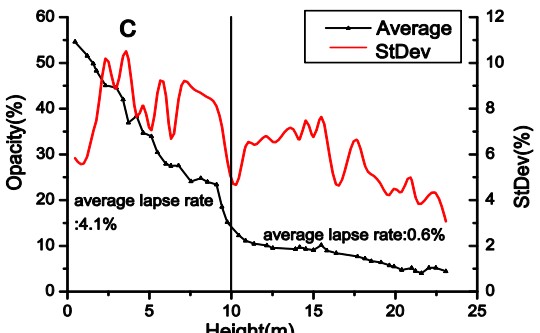

4  **Fig.3.** Vertical profiles of opacity of dust plume (a and b are the line-specific vertical profiles of

5  opacity of dust plume, c is averaged vertical profile and vertical standard deviations of opacity of

6  dust plume, the averaged lapse rate for the profile below 10m height is 4.1% over which it is 0.6%



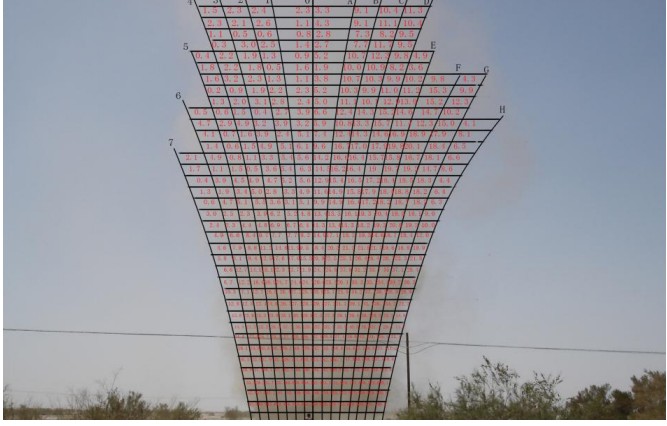

2          **Fig.4.**Horizontal variation of the DD's opacity(%) at different levels

5          **Fig.5.** Distribution of the DD's opacity in a vertical cross section





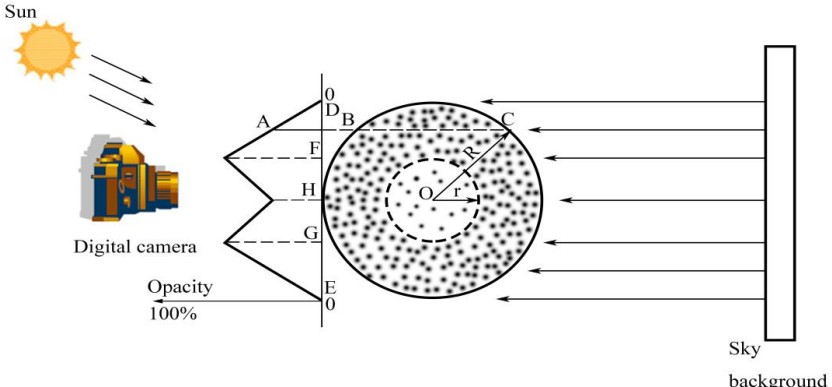

2          **Fig.6.** DD's horizontal section model and the corresponding variation of opacity

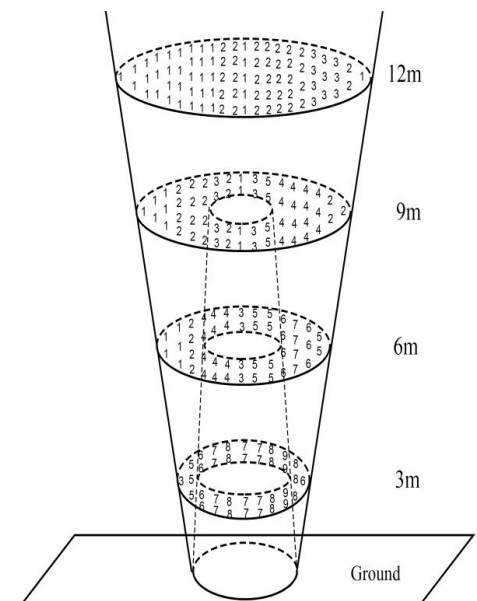

4          **Fig.7.** 3D structure of the DD's opacity

