# Peer review of "Dust Opacities inside Dust Devil Column in the Taklimakan Desert"

_Atmospheric Measurement Techniques, 2016_

## Referee Comment (RC1) · Anonymous Referee #3 · 18 Oct 2016

The authors did an excellent observational analysis on dust devil; however, further clarifications are needed to get the paper published:

1) The authors mentioned the structure of dust devil (DD)'s zone of weak winds is similar to the eye of tropical cyclone. However, according to figure 1b in the paper, the center of dust devil has strong upward motion, which is different from the hurricane eye (Figure 1R) or tornado center (Figure 2R) described in textbook, which has weak subsidence in their center. How to justify this inconsistency?

[Figure]

Figure1R schematic of hurricane structure from
https://upload.wikimedia.org/wikipedia/commons/thumb/4/4f/Hurricane-en.svg/800px-
Hurricane-en.svg.png

[Figure]

Figure 2R schematic of tornado structure from
https://www.britannica.com/media/full/599941/19397

2) It is true that the center of dust devil can has weak horizontal wind (e.g. the schematic drawing of dust devil from NASA website http://science.ksc.nasa.gov/mars/ops/dustdevil.gif ); and similar study (e.g., Zhang et. al. 2015) also confirmed that. However, no eye ( or subsidence) was claimed in all these previous studies. The reader of the paper may wonder whether the authors' current results are consistent with the results of previous studies.

[Figure]

Figure 3R schematic of dust devil

Reference mentioned in this review:

Zhang, M., X. Luo, T. Li, L. Zhang, X. Meng, K. Kase, S. Wada, C. W. Yu, and Z. Gu, 2016: From dust devil to sustainable swirling wind energy, Nature, 8322, doi:10.1038/srep08322.

---

## Author Comment (AC1) · 30 Oct 2016

The comment was uploaded in the form of a supplement: http://www.atmos-meas-tech-discuss.net/amt-2016-231/amt-2016-231-AC1-supplement.zip

---

## Referee Comment (RC2) · Anonymous Referee #4 · 30 Nov 2016

The authors present a technique to investigate the opacity of dust devils using digital imagery. Using this imagery, conclusions about the dust devil characteristics and structure are drawn.

While the presented method is interesting and promising for the investigation of dust devils, I see several shortcomings in the presentation and analysis of the results, which I think need to be accounted for before the manuscript can be considered for publication.

1) One major achievement presented in the manuscript seems to be the introduction of the Digital Optical Method (DOM) for the study of dust devils. The authors state that this is the first time this method has been used for this purpose. However, the recent paper by Liu et al. (2016, JGR), on which most of the authors are the same

as on the present paper, has already made use of this method. Perhaps the authors originally intended a different timing of the papers, but it would at least be important to mention the existence of a companion paper, which presents part of what the authors discuss as "Prospect" in Section 5. Given that the authors, location, and measurement technique are the same, I suppose that the time period of field observations should be the same, too (given as 7 − 14 July in Liu et al. and as 2 − 14 July in the present manuscript)? Please clarify.

2) Even though the authors state in Section 5 that "the results documented in this study are far from generalized characteristics of DD opacity", the authors draw very general conclusions throughout the manuscript, e.g. "The distinct horizontal distribution of opacity values proves the existence of the DD's eye" or also regarding the formation conditions and flow structure in dust devils. However, so far as I understand, the authors present results from only one example. How many dust devils have been observed/recorded during the 12-day observation period? I suppose more than one. Why are no statistics presented? The existence of a dust devil eye detected in one example does not necessarily mean that there has to be one in all dust devils. Understandably, the authors have selected a particularly well-structured dust devil to demonstrate the capabilities of their method. I think it is very important, however, to also show other cases, in which the dust devil structure is more complex, to see how the method performs under more difficult circumstances and to understand strengths and weaknesses of the method.

3) Several references used in the paper seem outdated (e.g. P. 2, L. 12) and some seem to be used in a misleading context. For example, it is stated that dust devils contribute 30% to global dust aerosol based on results from Koch and Renno (2005). A more recent study (Jemmett-Smith et al., 2015), suggests a much smaller percentage of about 3%. Other recent works support that dust devils likely contribute only ∼1% on large scale (see review of Klose et al., 2016, doi:10.1007/s11214-016-0261-4). Also, the statement that "30% of the primary particulate mass emitted to the atmosphere

in the Sahara Desert", which the authors attribute to Marsham et al. (2008), seems to be wrong. In my understanding, Marsham et al. (2008) conclude that the consideration of boundary-layer convection in their model increased dust uplift by 30% compared to an estimate using mean wind only. This does not mean that all of the 30% can be attributed to dust devils. I recommend a critical examination of the referencing, discussion of the reviewed estimates in more detail, and inclusion of newer references where available. I would like to refer the authors to an exhaustive review of dust devil studies published as a special issue by Space Science Reviews (http://link.springer.com/journal/11214/203/1/page/1).

4) Overall, the writing/language in the manuscript on hand needs to be improved. I am sure that some misleading statements can be related to difficulties in using English as the language for the paper.

5) P. 1, L. 14 and P. 2, L. 10: "vortexes" instead of "vortexes of wind".

6) P. 1, L. 14: Please provide a reference for the given dust devil height-range of 1 – 1000m

7) P. 1, L. 19: "swirling vortex" seems to be a tautology.

8) P. 2, L. 13 – 19: While weak winds and sunny weather might lead to dust devil formation, pressure fluctuations are a consequence, not a cause, of dust devil occurrence. Please clarify. Also, the subsequent discussion of heterogeneous solar radiation is somewhat confusing, and "certain conditions of angular momentum" seems very vague. Please revise.

9) P. 3, L.7 – 11: While I know that dust particles can serve as ice nuclei and cloud condensation nuclei, and that dust can supply iron for oceanic phytoplankton growth, I do not know an "umbrella affect". It might be beneficial to briefly outline all three effects. Further, the effects are known to apply for dust aerosol in general, but I am not aware that the relevance of dust devils in the context is known or has been investigated at all.

10) P. 3, L. 12: I recommend referring to Neakrase et al. (2010, doi: 10.1007/s11214-016-0296-6) for a discussion of particle lifting processes in dust devils

11) P. 3, L. 18 – 22: The authors seem to suggest that, compared to their technique, one disadvantage of the use of Doppler radar and pressure loggers for the study of dust devils is that the instruments need to be deployed in the estimated path of a dust devil. While this is certainly the case, I think the same applies to the use of a fixed camera, doesn't it? I also think that the cost of the pressure loggers as described by Lorenz (2012) can with approx. \$120 not referred to as high.

12) P. 4, L. 2 – 3: So far as I know, Kanak et al. (2000, 2005) conducted simulations with zero wind, so can probably not confirm neither deny that weaker winds and stronger surface heat fluxes are favorable for dust devil formation. See also Klose and Shao (2016, doi: 10.1016/j.aeolia.2016.05.003) for a study of the dependence of dust devil formation on atmospheric conditions using LES, Spiga et al. (2016) for a review on LES used for dust devil studies, and Rafkin et al. (2016) for a review on dust devil formation conditions (the latter two references can be found in link within comment 3).

13) P. 5, L. 21: How is dust devil height determined? Is the "upper end" of a dust devil defined by a particular opacity threshold?

14) P. 6, L. 1 – 2: Why are the lines chosen to follow a conical pattern rather than a regular grid? I could imagine that in particular for dust devils that are not well-structured and might deviate substantially from an ideal conical shape, the choice of a dust-devil-shaped grid is problematic. A regular grid might also potentially enable a better comparison between differently shaped dust devils.

15) P. 6, L. 16 – 22: Is this derivation needed if the result is N = N0 for a plume-free atmosphere?

16) Figure 3: I think it would be easier to compare the different lines if they were all in the same figure. Also, Figure 3c does not seem to contain much additional information,

so I suggest removing the figure.

17) Figures 5 and 7: I suggest using colors rather than numbers to visualize the opacity patterns.

18) P. 9, L. 8 – 10: Another reason for an expected opacity decrease with height is probably that the dust source is at the bottom and a higher particle concentration is expected close to the source. While gravitational settling certainly leads to a strong decrease of larger-sized particles with height, smaller particles might actually create higher opacity than larger particles (at comparable concentrations). What do the authors think how this would affect the vertical opacity profile?

19) P. 12, L.22 – P. 13, L. 2: The fact that the opacity in a dust devil depends on its formation conditions and intensity seems clear. The investigation of more examples may already provide more insights in the variability of dust devil opacity. This might also be a good opportunity to reference to the companion paper of Liu et al. (2016), in which dust devil formation conditions are investigated.
* * *

---

## Author Comment (AC2) · 24 Dec 2016

The comment was uploaded in the form of a supplement: http://www.atmos-meas-tech-discuss.net/amt-2016-231/amt-2016-231-AC2-supplement.zip
* * *

---

## Author Comment (AC3) · 27 Dec 2016

The comment was uploaded in the form of a supplement:
http://www.atmos-meas-tech-discuss.net/amt-2016-231/amt-2016-231-AC3-supplement.zip
* * *